# The Effects of Combined Movement and Storytelling Intervention on Motor Skills in South Asian and White Children Aged 5–6 Years Living in the United Kingdom

**DOI:** 10.3390/ijerph17103391

**Published:** 2020-05-13

**Authors:** Emma L. J. Eyre, Cain C. T. Clark, Jason Tallis, Danielle Hodson, Sean Lowton-Smith, Charlotte Nelson, Mark Noon, Michael J. Duncan

**Affiliations:** 1Centre for Sport, Exercise and Life Sciences, Coventry University, Coventry CF4W + VG, UK; ad0183@coventry.ac.uk (C.C.T.C.); Jason.tallis@coventry.ac.uk (J.T.); nelson12@uni.coventry.ac.uk (C.N.); Mark.noon@coventry.ac.uk (M.N.); Michael.duncan@coventry.ac.uk (M.J.D.); 2School of Social and Health Sciences, Sport, Health and Physical Educatioon, Leeds Trinity University, Horsforth, Leeds LS18 5HD, UK; d.hodson@leedstrinity.ac.uk; 3School of Human Sciences, University of Derby, Derby DE22 1GB, UK; S.lowton-smith@derby.ac.uk

**Keywords:** motor skill instruction, fundamental movement skills, ethnicity, disadvantaged, locomotor, object control

## Abstract

Early motor development has an important role in promoting physical activity (PA) during childhood and across the lifespan. Children from South Asian backgrounds are less active and have poorer motor skills, thus identifying the need for early motor skill instruction. This study examines the effect of a movement and storytelling intervention on South Asian children’s motor skills. Following ethics approval and consent, 39 children (46% South Asian) participated in a 12-week movement and storytelling intervention. Pre and post, seven motor skills (run, jump, throw, catch, stationary dribble, roll, and kick) were assessed using Children’s Activity and Movement in Preschool Study protocol. At baseline, South Asian children had poorer performance of motor skills. Following the intervention, all children improved their motor skills, with a bigger improvement observed for South Asian children. Early intervention provided remedial benefits to delays in motor skills and narrowed the motor skills gap in ethnic groups.

## 1. Introduction

The development of fundamental movement skills (FMS) in childhood is considered a key driver for lifelong physical activity (PA), weight management, and fitness [1]. FMS, which include object control, locomotion, and body management skills (e.g., balance), are the basic motor skills that lead to the development of specialised sports skills [2]. Children have the developmental ability to master most FMS by the age of six years [2]; as such, these skills are a key focus in the early years’ national curriculum [3]. Despite the key focus and developmental potential, globally, large numbers of children fail to master these [4,5,6], and thus a proficiency barrier is seen. Understanding effective ways to overcome such barriers is a key step for public health professionals looking to enhance PA and movement in educational and community health care settings.

These proficiency barriers may be more pronounced for specific groups (e.g., children with development coordination delay, disadvantaged children). Surprisingly, despite the ethnically diverse and multicultural societies represented worldwide, few research studies have examined ethnicity as a determinant of FMS [7]. Specifically, a group of concern are people from South Asian backgrounds (i.e., Indian, Pakistani, or Bangladeshi backgrounds) living in the United Kingdom. South Asians have higher body fat, increased risk of cardio-metabolic disease [8,9,10], and lower PA levels [11,12,13,14] in comparison with white children/adults. Hence, these health differences may in part be explained by the failure to develop FMS in childhood. Considering Stodden et al.’s [15] development trajectory model, failure to develop FMS would negatively impact on PA levels and healthy weight maintenance. Preliminary findings into ethnic differences in motor skills suggest these ethnic groups at higher health risk may have poorer motor skills [16,17,18,19]. These health and motor delay risks may be further complicated by the association between ethnicity and socio-economic status, whereby people from South Asian backgrounds are the most socio-economically deprived [20,21]. Children from deprived backgrounds are also at risk of having development delays [22,23,24], lower PA [25], and thus poorer health. Considering Newell’s constraint theory [26], poorer development of motor skills in these populations may be influenced by the constraints (i.e., individual, task, environment) those children are exposed to, which, in turn, delays the development of these skills. To date, no study has considered this in the context of children from different ethnic backgrounds living in the United Kingdom. This highlights the need for early intervention in these at-risk groups (i.e., ethnic, deprived, and low FMS) to remediate development delays.

In order to develop motor skill proficiency, skills must be developed through appropriate structure [27] and developmentally appropriate activities and feedback within learning environments [28]. School- and community-based programs that include developmentally appropriate FMS learning experiences delivered by specialists or highly trained classroom teachers have significantly improved FMS proficiency levels in children [29]. Significant improvements in FMS following a 12-week motor skill intervention have also been found in disadvantaged and developmentally delayed preschoolers [27], evidencing that those who may have developmental delays can improve from such focused interventions. Interventions have been most successful when they have included either direct instruction, parent-assisted, or student-centered approaches, with improvements reported following eight weeks’ intervention [27]. Effective instruction is considered best practice to enable individuals to develop at their own rate and includes practice appropriate to the learning goal, providing a variation in tasks to enable learner success and clear communication of tasks/outcomes to individuals [30]. Despite the many interventions employed to improve FMS [29], little research has been conducted in children from ethnic backgrounds, and thus little is known about how to develop these skills within this population. It has been proposed that language and motor skills are related and, when language ability is impaired, a motor deficit is observed [31]. Given that that these are key development areas in the U.K. statuary framework for the Early Years Foundation Stage [3], an approach that combines these opportunities is of value. This is particularly the case as evidence suggests all children are performing poorly in these domains [32]. Given that children from ethnic backgrounds are less likely to engage in PA [11,12], more likely to have English as additional language, and less likely to have role models for PA outside of the school [33,34], providing effective interventions to narrow inequalities in future health is needed.

There is some evidence to show that combining movement and academic activities (i.e., storytelling) results in superior motor development and language ability compared with movement or storytelling alone [35]. This is considered an embodied cognitive approach, whereby sensorimotor experiences are gained through bodily actions within the environment, which are important for developing cognitive, creative, and motor outcomes, particularly in early childhood [36]. The focus on integrating story with movement potentially provides a more creative stimulus than movement only related intervention designs. With such an approach, the interaction of tactile input, social input, via peer interaction, with sensorimotor activity in a creative manner has been suggested to drive changes in executive functioning and motor skill in children [37]. In an embodied paradigm, performing actions/movements leads to the construction of rich and elaborative representations in the brain, which enhance memory recall. The “enactment effect” engages the motor system, in which encoding is facilitated compared with just observing the same events [38]. It is this enactment effect that enables the intervention to be successful. This is an approach that can be more easily accommodated in a crowded curriculum and provides a mechanism for embedding learning objectives across the curriculum. However, no research has examined this as an approach in ethnic children living in deprived areas. This is particularly important as these aforementioned groups experience poorer motor and language skills and could benefit more from such an intervention than their white, English as a first language, and less deprived peers. Given that the development of these skills will not occur naturally [39], without motor skill instruction and appropriate structure and feedback [27,40,41], any intervention that narrows inequalities in motor skill and language differences in children from ethnic minority groups would be of value to educators.

The current study sought to be the first study to (1) compare baseline data on FMS between South Asian and white children matched by age and deprivation; and (2) to determine the influence of a movement and storytelling intervention on the development of motor skills and their associated movement process characteristics in South Asian and white children matched by age and deprivation.

## 2. Materials and Methods

A 12-week movement and storytelling intervention in children from South Asian backgrounds and those from white backgrounds was employed. Motor skill performance was assessed at baseline and at the end of the 12-week intervention period using process- and product-based assessments. The inclusion of both process and product is considered to provide a more holistic overview of FMS changes [42], yet, to date, few studies use both.

Participants for the motor skill intervention were recruited from school entry age (five years) using cluster sampling at ward and school level in central England, following institutional ethics approval (P38406), parental informed consent, and child assent. A quasi-randomised design, matched by low deprivation using the Index of Multiple Deprivation [43], was employed, given that the study was undertaken in a natural setting of primary schools. In this instance, white children from the same socio-economic backgrounds were used as a matched control. This is because the intervention was previously shown to be effective in white children predominantly in comparison with the normal Physical Education curriculum delivery [35]. Therefore, it was deemed pragmatic to compare the differential effects of the intervention on the target population (i.e., South Asian) to a group where the effects were already shown (i.e., white). A total of 87 children were recruited in the study, with 39 completing all of the data collection and motor skill sessions (46% South Asian intervention group). Children’s ethnic background was determined using school records in accordance with the Department for Education and Skills.

### 2.1. Procedure

Children were assessed in the school setting. Height (cm) and mass (kg) were assessed, with children in bare feet and wearing light shorts and t-shirt, using a SECA stadiometre and weighing scales (SECA Instruments Ltd., Hamburg, Germany). Actual motor competence (process and product measures) was assessed with children in small groups. These assessments were conducted at baseline and the end of the motor skill intervention. All testing took place in the same setting, same time of day, and in the same sports hall to control for any external factors influencing the findings.

#### 2.1.1. Process Measures of Motor Skill Performance

Process movement assessments are concerned with how a skill is performed (e.g., technique, sequencing of movement) [44]. Children’s Activity and Movement in Preschool Study (CMSP) motor skills protocol [45] was used to assess process movement proficiency in seven motor skills, two locomotor (run and jump) and five object control skills (overarm throw, catch, stationary dribble, kick, underhand roll) at baseline and post intervention. The CMSP is a motor skill assessment tool developed from the test of gross motor development (TGMD-2), which is more suitable for pre-school aged children [43]. Construct and concurrent validity (r = 0.94–0.98) and reliability (r = 0.88–0.97) have been shown for this age group [45]. The child’s performance of each trial for each skill was video-recorded (Casio, EX-F1, Tokyo, Japan) in the sagittal plane, and subsequently edited into single film clips of individual skills on a computer using Quintic Biomechanics analysis software V21 (Quintic Consultancy Ltd., Birmingham, UK). These files were scored on their movement process characteristics according to Williams et al. [45]. Two researchers experienced in the assessment of children’s movement skills (having previously assessed movement skills in the context of a previous research study) analysed the videos. Both raters were previously trained in two separate two to three hour sessions. Raters watched videos of children’s motor skill performances and rated these against a previously rated ‘gold standard’ rating. Congruent with prior research [46], training was considered complete when each observer’s scores for the two trials differed by no more than one unit from the instructor score for each skill (>80% agreement). Inter- and intra-rater reliability analysis was performed for all the skills between the two researchers. Inter-rater reliability was 81% and intra-rater reliability was 87%, demonstrating good reliability [47,48,49]. 

Scores on the CMSP were ratings of movement process characteristics of each individual skill as well as skills in the locomotor subscale, object control subscale, and total test categories (i.e., the sum of all seven skills). Each skill comprises five to seven movement process characteristics (also referred to as behaviour components) listed in the CMSP. Each skill was scored out of two attempts using these individual movement process characteristics. Movement process characteristics were rated as “1” (present) or “0” (not present) for most skills and summed (separately for 2) to arrive at a score for locomotor, object control, and total test performances. Exceptions to the rating of “1” and “0” were made for one process characteristic of the throw. For the throw, “hip-trunk rotation” was scored as “2” (differentiated), “1” (block), and “0” (no rotation) [43]. Correct performance was determined by the evidence of/absence of process characteristics of each skill presented in the guidelines for scoring using the CMSP protocol [45]. Scores from two trials were summed to create a total for each skill (scored 0–24 for locomotor, 0–62 for object control, and 0–86 for total FMS).

Individual process components of the skills were also grouped as ‘mastery’, which is correct performance of skill process components on both trials; ‘near mastery’, which is correct performance in one of both trials; and ‘poor’, which is any score below the above in accordance of the work of prior research [49,50] based on process characteristics defined in CMSP.

#### 2.1.2. Product Measures of Motor Skill Performance Competence

Product measures are movements assessed as an outcome or product of skill executions (e.g., distance jumped or thrown, time taken to complete a run) [44]. Three product measures—10 m sprint time, standing long jump, and medicine ball throw—were assessed following one demonstration by the researcher. A 10 m sprint run was timed using smart speed gates (Fusion Sport, Coopers Plains, Australia) and standing long jump measured (distance from the take-off line to the back of the closest heel on landing) using a tape measure. Two trials were used, with the fastest time and longest jump being used for analysis. Both assessments have demonstrated acceptable test–retest reliability (ICC = 0.81–0.940) [51], in children aged 5–9 years. For the medicine ball throw assessment, the protocol described in Davis et al. [52] was followed and an average of three trials were computed, which is deemed to be a reliable (ICC = 0.93–0.94) and valid assessment in children aged 5–7 years [52].

### 2.2. Movement and Storytelling Intervention

The intervention was carried out by the principal investigator at all times alongside the classroom teaching assistant, to provide an environment similar to that which would be translatable following the intervention. All sessions were undertaken during the indoor school hall based on a group size of 25–30, which is consistent with class sizes for Physical Education. Following the baseline assessments, all groups undertook the same 12-week movement and storytelling (time = 35 min, once per week). 

The intervention built upon the prior work of Duncan et al. [36], which had previously been successful in enhancing a small number of motor skills (run, jump, throw, and catch) in a sample of white preschoolers. Each week, the children would undertake movements related to characters from popular children’s story books (‘The Gruffalo’ and ‘Stickman’ [53]). Each session lasted for 35 min and comprised of the following: (a) warm up and introduction (6 min); (b) skill stations (3 × 6 min with children rotating stations rotated) or whole group activity (2 × 9 min); and (c) cool down and closure of skill instruction (6 min, Table 1). The remaining 5 min were allocated for transition between activities. When stations were used, children worked in groups of 5–6 (which rotated) and were grouped according to academic groups. When children were involved in whole group activities, these were either carried out individually, in pairs, or in groups of the three. A summary of the approach can be found in Table 1 and a detailed breakdown of these activities can be found in the movement and storytelling manual [54]. Each session was focused on teaching one FMS, with a progression of 2–3 instructional activities, which were developmentally appropriate. The instructor presented the task and provided a demonstration for each activity, and key words were reinforced throughout the activities that related to critical elements of the skill. Children received instruction and feedback by either the lead researcher or the classroom teaching assistant, and this was positive specific and positive corrective feedback. These plans were designed in accordance with the work of Goodway et al. [27], whereby this design previously resulted in increases in motor proficiency in disadvantaged children. Prior to all lessons, the lead researcher and the classroom teaching assistant would go over the lesson plan and discuss key instruction elements. In the closure of each session, children would be shown an image from the book and asked to describe something about a particular character in the book as well as questions related to motor skill execution (e.g., how does the mouse move? Who does the mouse meet? Where does the fox live? What does gruffalo look like? Why did the fox run away from the mouse? Where did your foot finish when you kicked the ball? How can you jump the furthest?). This process was included to better embed the movements involved in each session with the aspects of the story for each particular session.

### 2.3. Data Analysis

In order to estimate sample size, we used a mean expected difference of 0.5 with a generous standard deviation of 2.0, an alpha error probability of 0.05, and a power (1-beta error probability) of 0.95, resulting in 36 participants being required. We examined between and within-subject differences, pre and post intervention, using pairwise Tukey’s honestly significant difference (HSD) tests (thus correcting the alpha level for multiple comparisons), in and reported alongside corresponding effect sizes (Cohen’s d, classified as small (0.2), medium (0.5), large (0.8), or very large (1.3) [55,56]; and 95% confidence intervals (CIs)). In line with recommendations of Barnett et al. [57], to reduce the potential impact of the regression to the mean phenomenon, follow-up measurements were conducted in all participants, and measurement error was reduced by using assessments with high construct and concurrent validity (r = 0.94–0.98) and reliability (r = 0.88–0.97) [47].

Bayes factors were also computed to express the probability of a difference given H_10_ (alternate hypothesis) relative to H_01_ (null hypothesis; that is, values larger than 1 are in favour of H_01_), assuming that H_01_ and H_10_ are equally likely, and using a default prior [58,59,60]. Bayes factors were reported as the probability of the data given the alternate, relative to the null hypothesis, or vice-versa (classified as anecdotal (_BF_1–3), moderate (_BF_3–10), strong (_BF_10–30), very strong (_BF_30–100), or extreme (_BF_ > 100)) [58,59,60]. Bayesian analysis was incorporated because it allows the combination of domain-specific knowledge, permits direct probability statements to be made about parameters (population level effects), allows zero effects to be determined, provides estimates of uncertainty around parameter values that are more intuitively interpretable than those from null hypothesis testing alone, and aids in the interpretation of *p*-values [61,62]. A chi square test was used to assess independence of ethnic grouping by component level of the skill and, where findings were significant, Cramer’s V was used. The effect was determined as small if V = 0.1, medium if V = 0.30, or large if V = 0.50 [63]. All analyses were conducted in R [64], using the bayesfactor package [65], and jamovi software extension (The jamovi project, 2019). jamovi. (Version 1.1.7) [Computer Software]. Retrieved from https://www.jamovi.org).

## 3. Results

### 3.1. Ethnic Comparison of Motor Skill Performance at Baseline

Descriptive data for participant characteristics are detailed in Table 2, and there were no significant differences between ethnicities for age, height, sitting height, weight, body mass index (BMI), and waist circumference (WC) (all p > 0.05). Overall skill and individual components data are presented in Table 3, 4, and 5, respectively. We found that, at baseline, there were significant differences between ethnicities for run (white > South Asian, p = 0.01, d = 0.51, BF: 3.28; moderate evidence of difference), stationary dribble (white > South Asian, p = 0.03, d = 0.41, BF: 1.42; anecdotal evidence of difference), throw (white > South Asian, p = 0.0002, d = 0.83, BF: 133; extreme evidence of difference), roll (white > South Asian, p = 0.004, d = 0.62, BF: 8.76; moderate evidence of difference), 7-skills score (white > South Asian, p = 0.002, d = 0.71, BF: 14.4; moderate evidence of difference), and medicine ball throw (white > South Asian, p = 0.004, d = 0.64, BF: 8.6; moderate evidence of difference) (Table 3). At baseline, the significant test for independence indicated an association between ethnicity and skill level with medium to large effects for run component 1; jump component 2; stationary dribble component 1 and 5; kick component 2 and 3; and roll component 1, 2, and 5 (Table 4). In these instances, a larger proportion of South Asian children were categorised as poor in motor performance of the skill component and/or higher levels of white children showing mastery of the motor component. Yet, South Asian children had higher proportions able to demonstrate component 4 and 6 of the jump (Table 5).

### 3.2. Ethnic Comparison of Motor Skill Performance Following the Intervention

At intervention cessation, we found there were significant differences between ethnicities for kick (South Asian > white, p < 0.0001, d = 1.68, BF: 1.49^8^; extreme evidence of difference), throw (South Asian > white, p < 0.0001, d = 1.29, BF: 85,660; extreme evidence of difference), roll (South Asian > white, p = 0.0008, d = 0.69, BF: 37.5; very strong evidence of difference), 7-skills score (South Asian > white, p < 0.0001, d = 2.06, BF: 3.68^9^; extreme evidence of difference), long jump (white > South Asian, p < 0.0001, d = 0.95, BF: 1021; extreme evidence of difference), and medicine ball throw (white > South Asian, p < 0.0001, d = 1.23, BF: 61,767; extreme evidence of difference). Furthermore, significantly higher proportions of children from South Asian backgrounds were able to demonstrate the following components of the skill (i.e., stationary dribble component 1, 3, and 4; kick component 2, 4, and 5; overarm throw skill component 2, 5, and 6; roll component 1, 2, 4, 5, and 6; and catch component 6; Table 5).

### 3.3. Motor Skill Performance Change from Baseline to Post Intervention

When considering the changes pre to post intervention, no significant within or between-subject differences were found for locomotor total, object total or run, throw, jump, catch total. However, a significant difference between pre-7-skills to post-7-skills score for the white (p < 0.0001; mean difference: 11.89; 95% CI: 7.81–15.97; d = 1.13; BF: 8573, extreme evidence of difference) and South Asian (p < 0.0001; mean difference: 33.13; 95% CI: 30.49–35.78; d = 5.54; BF: 2.23^14^, extreme evidence of difference) ethnic groups, respectively. A bigger change pre to post was observed for South Asian children. Descriptive statistics by all skills, individual skills, and ethnic grouping pre to post can be found in Table 3 and Table 4.

#### 3.3.1. Individual Locomotor Skills: Process Measures

There was a significant difference between pre-run to post-run score for the South Asian ethnic group only (p < 0.001; mean difference: 2.46; 95% CI: 1.29–3.63; d = 0.68; BF: 191, extreme evidence of difference). No significant between-subject differences were found for run or jump and no within differences for jump.

#### 3.3.2. Individual Object Control Skills: Process Measures

South Asian children significantly improved pre to post for stationary dribble (p < 0.001; mean difference: 4.52; 95% CI: 3.59–5.46; d = 1.69; BF: 4.15^8^, extreme evidence of difference) and overarm throw (p < 0.0001; mean difference: 4.68; 95% CI: 3.8–5.58; d = 2.01; BF: 5.66^8^, extreme evidence of difference). Both white and South Asian children significantly improved their kick from pre to post (white; p = 0.01; mean difference: 1.63; 95% CI: 0.58–2.69; d = 0.52; BF: 11.2, strong evidence of difference and South Asian; p < 0.0001; mean difference: 6.78; 95% CI: 5.83–7.74; d = 2.56; BF: 2.58^12^, extreme evidence of difference). A significantly greater improvement in the South Asian versus white ethnic group (p = 0.007; BF: 1.99, anecdotal evidence of difference) was seen. For white children, a significant difference between pre-roll to post-roll score for the white ethnic group only (p < 0.0001; mean difference: −3.75; 95% CI: −4.99–2.5; d = 0.96; BF: 42463, extreme evidence of difference) was found. No significant between-subject differencse were found for stationary dribble, overarm throw, and underarm throw, roll, and catch. No significant between subject differences were found for catch.

#### 3.3.3. Product Measures of Motor Skill Performance

There were no significant within subject differences for sprint speed or medicine ball throwing distance. No between subject differences were found for sprint speed or jump distance. However, white children jumped significantly further post intervention compared with pre (p < 0.0001; mean difference: 20.06; 95% CI: 11.85–28.27; d = 0.74; BF: 1567, extreme evidence of difference). There was a significantly greater improvement in medicine ball throwing distance in the white versus South Asian ethnic group (p < 0.0001; BF: 12,328, extreme evidence of difference).

## 4. Discussion

The current study is unique in examining differential effectiveness of a school-based movement skill intervention on white and South Asian children, the latter being a group recognised as less active, at risk of poor health owing to inactivity, and delayed motor patterns [11,12,16]. Importantly, this study is one of only a few examining motor skills in ethnic groups. Additionally, this is the first focusing on ethnic differences to present changes in motor skill at individual skill and movement process characteristic, and to incorporate Bayesian statistical perspectives, alongside traditional frequentist approaches.

### 4.1. Baseline Fundamental Motor Skill 

The current study identifies that, at baseline, young South Asian children living in deprived areas have poorer motor skill development than white children matched by age and deprivation. This is the first study to establish, at a skill level and movement process level, where motor skill differences exist. Specifically, these ethnic differences were found for run (move their arms in opposition to legs with their elbows bent when running), overarm throw (initiate a wind up by a downward movement of hand/arm when throwing the ball overarm), stationary dribble (move their arm independent of trunk and control the ball for four consecutive bounces when bouncing the ball), rolling a ball (ball arm/arm swings down and back of trunk; chest/head face forward, arm action in vertical plane and ball held in fingertips when rolling the ball), as well as medicine ball throwing distance. In all instances, more white children were able to display these components. It is difficult to compare our findings to previous studies because other studies have not examined skill differences at a movement process characteristics level in children. Of the sparse data that do exist comparing motor skills in ethnic groups, our ethnic differences found build on prior findings that identified poorer locomotor skills in South Asian children living in the United Kingdom (age = 5 years) [16] and poorer object control skills in Asian–Australian children (age = 9–11 years) [19]; thus, motor development delays in South Asian children are observed. These differences may be explained by Newell’s constraint theory [26], which identifies task constraints. Physical factors (e.g., physical fitness) and mechanical factors (e.g., stability factors, giving force factors, and receiving force factors) are known to affect the ability to complete a motor task [2]. In this instance, the delayed motor patterns seen in South Asian children may indicate poorer mechanical efficiency of the arms and force production. This is because good neuromotor development underpins the mechanical ability to execute motor skills and the same neuromotor development is linked to muscular strength and endurance [15]. This mechanical ability is developed through appropriately structured teaching and learning strategies as a consequence of activity in school physical education, engagement in active play with parents, and taking part in community sports activities. The ability to wind up the hand/arm prior to overarm throwing relies on a combination of mechanical efficiency, strength, and neural control to actually coordinate the limbs in a rhythmical pattern and exert force to propel the ball. There is some evidence to suggest that South Asian children may have poorer fitness and physical activity, and are born lighter with more preserved fat mass [9,10,11,12,66], all which impact on motor development, but limited data exist on the neuromuscular control in South Asian children. This is an area that warrants further investigation. The fact that there are differences in ability to execute particular process characteristics of some skills between white and South Asian children is important as it identifies aspects of movement performance that may need specific focus on interventions with South Asian children. It would suggest, in the current case, that either the white children engage in greater community sport participation or they may engage in more active play focused on motor skill than their South Asian peers. This is speculative and additional research is needed to substantiate these claims.

Contrastingly, both groups performed similarly for jumping, catching, and kicking. Similarly, all children were best at running, catching, kicking, and rolling and poorest at jumping, throwing, and stationary dribble. It is not surprising that the score for all children is highest for running, as running is one of the earliest emerging motor skills [2]. Given that jumping is a body projection skill requiring muscular strength, multi-limb coordination, and dynamic balance to perform proficiently, and is one of the skills that takes the longest to develop from the emergent to proficient phase, it is not surprising that this skill was one of the poorest in all of the children [66]. Additionally, throwing is also a complex motor skill that requires the interactions of different body parts to coordinate with each other in applying biomechanical principles of action/reaction (for example, the use of stepping and rotating different body parts in sequence to transfer forces); again, throwing is one of the skills that has the longest timeframe of development [67].

### 4.2. The Influence of Movement and Storytelling Intervention on All Fundamental Motor Skills

This is the first study to examine the effects of combining movement and storytelling in a deprived South Asian group during early childhood, a group at risk of motor delay. The findings identify that, despite being delayed in motor skills at baseline in comparison with white age and deprivation matched children, a movement and storytelling intervention can result in improved motor proficiency for both ethnic groups, but that South Asian children gained more from the intervention, thus narrowing the motor skill gap. This highlights the importance of early intervention in reducing the proficiency delays seen in this population, as well as further supporting the use of motor skill interventions for effective motor skill development [28,29,35]. This also supports the preliminary evidence identifying the success of combining movement and storytelling for motor development [35] and the need for bespoke interventions in early years settings [68,69]. Given the intervention success with a small dose (35 min), the intervention type may be of interest to school practitioners who need to deliver and maximise benefits of their intervening in a crowded curriculum. The increased improvement seen in the South Asian group may be a consequence of several factors, firstly, these children were more delayed at baseline, and thus had more room for improvement. Furthermore, differences may be explained by Newell [26] constraint theory, whereby constraints were structured in a way that was facilitative to motor development. However, what is apparent and not identified in other motor skill interventions, is the variation in the responses to the intervention at an individual level. This highlights, firstly, that some may get differentiated benefits; secondly, that the outcome of interests are variable given that, at this age skills are being learnt, this is not surprising given that great variation in developmental patterns across age. In combination with manipulating the constraints from Newell’s model [26], the combination of movement and storytelling through the role of embodied movement should also be considered [32,37,38]. In the present study, it is possible that the sensorimotor experiences provided as part of the movement intervention, coupled with the focus on storytelling, serve to augment the changes in motor skills for the children who engage in this intervention. This is an approach that can be more easily accommodated in the curriculum and the use of story may provide more interest in the activities.

### 4.3. The Influence of Movement and Storytelling Intervention on Component Level of Fundamental Motor Skill

Despite being poorer at baseline, following the intervention, South Asian children had higher scores for kicking, throwing, and rolling than white children. These differences were specifically seen in preparation stages of throwing and rolling (component 1 and 2), and force production stage and follow through of rolling (component 4, 5, and 6) and kicking (component 2, 4, and 5). The force production stage of throwing was better for white children, with more being able to demonstrate differentiated rotation. For South Asian children, block rotation was used for force production. Independent of body size, white children were able to throw the medicine ball further and jump further following the intervention. Given the use of product and process scores in this study, further jumping distances generated in white children may explain why more children from these backgrounds landed non simultaneously in comparison with South Asian children. While this provides insight, it is difficult to make further interpretations or conclusions on the finding. This is because a lack of comprehensive understanding exists in the relationship between process and product assessments of motor skills in young children [40], thus such differences need to be interpreted with caution.

### 4.4. Limitations

There are several limitations to consider in the work. Firstly, a retention rate of 44.8% was achieved, and thus we do not know how effective the intervention was for those who were not retained. This was because, to meet the inclusion criteria, each child needed to have attended weekly and both assessment periods. This was to enable us to ensure that the all achieved a similar intervention dose. Those that did not attend all sessions were because of common seasonal illness (e.g., virus, colds), preventing them from attending school that day. While this retention rate is low, it is representative of the age of the children and the attendance figures. In future work, the examination of minimum doses may enable further insight into how much dose is needed to be effective. Secondly, the work compares children from schools from similar deprived environments (lowest 10%) to attempt to minimise the influence of socio-economic status (SES). This approach was taken because it was pragmatic owing to the inability to obtain individual SES data. However, this means that the children were not directly matched individually, and thus some children may have been more deprived than others. In future studies, the inclusion of these individual data would prove useful. Thirdly, it was not possible to obtain language and literacy metrics in a reliable and comparable way, given that these are assessed differently at these schools. Collecting these metrics in future studies alongside motor skill, as well as the follow up of these outcomes, would provide a useful next step for researchers. Fourthly, while we are aware of the impact of the intervention, it is difficult to ascertain why these differences occurred, particularly why some gained more than others. Further examination of these developmental sequence changes with the use of 3D motion capture in understanding mechanical principles from in-depth kinetic and kinematic perspectives in addition to the exploration of physical factors may help to explain these differences in future work. Finally, Bayesian analysis was incorporated because it allows the incorporation of domain-specific knowledge and permits direct probability statements to be made about parameters, which aids in the interpretation of p-values [58,59]. As such, the current study makes an original contribution to the literature base related to motor skill. In accordance with Barnett et al., we actively sought to reduce the potential impact of the regression to the mean phenomenon by ensuring follow-up measurements were conducted in all participants, and measurement error was reduced by using assessments with high construct and concurrent validity (r = 0.94–0.98) and reliability (r = 0.88–0.97) [43]. However, we cannot disregard the potential for residual confounding owing to the lack of the presence of a control group. Thus, we suggest that, given the positive findings reported here, further work should be conducted following a randomized controlled trial study design.

## 5. Conclusions

The study makes a unique contribution by demonstrating low levels of motor skill performance in young children and particularly those from South Asian backgrounds. Furthermore, the study identifies the importance of early intervention in these groups in providing remedial benefits to such delays. Combining movement and storytelling narrowed the motor skill gap in South Asian children. Further research needs to understand the mechanisms for such changes including obtaining language, mechanical, and physical factor data. Exploring the sustainability of these factors over time warrants further work.

## Figures and Tables

**Table 1 ijerph-17-03391-t001:** Example of the movement and storytelling programme for two weeks.

Week/Focus	Opening (6 min)	Skill Station/Activity (18 min)	Closing Activity (6 min)	Demonstration with Key Words and Transition Time (5 min)
Week 2: Gruffalo story	Story passage and questioning	Branch drop (whole group)	Adventure back from the forest to bed with questioning related to the movements and book extract.	Step-hop Swing arms Eyes watching
Owl is skipping	Adventure to the forest	Musical owls (whole group)		
Week 7: catching Stickman	Story passage and questioning	Fetch and drop (whole group)	Bean game in reverse speed with questioning related to the movements and book extract.	Hands and eyes ready Reach for ball Pull to chest Two hands
Stickman is used for fetch and drop	Bean game and mobility exercises	Stick catch with friends (whole group		

**Table 2 ijerph-17-03391-t002:** Participant characteristics: mean (SD).

Ethnicity	Age	Height (cm)	Sitting Height (cm)	Weight (kg)	BMI (kg/m^2^)	WC (cm)
White	5.06	108	56.42	18.78	15.98	52.62
	(0.28)	(5.14)	(1.497)	(2.184)	(1.131)	(3.637)
South Asian	4.61	112	55.23	19.2	15.34	49.54
	(1.18)	(8.42)	(3.083)	(3.93)	(1.789)	(4.608)

WC, waist circumference; BMI, body mass index.

**Table 3 ijerph-17-03391-t003:** Overall skills pre vs. post: mean (SD).

	Locomotor			Object			Run, Jump, Throw, Catch			7 Skills		
Ethnicity	Pre	Post	∆	Pre	Post	∆	Pre	Post	∆	Pre	Post	∆
White	15	18	3.55	28	31	2.94	26	33	6.67	49 ^1^	52	11.3 *
	(5)	(4)		(13)	(11)		(9)	(7)		(8)	(9)	
South Asian	14	17	2.89	27	34	6.50	26	31	5.94	43	68 ^2^	32.96 *
	(4)	(6)		(5)	(22)		(6)	(9)		(7)	(5)	

* denotes *p* < 0.0001; ∆ denotes mean difference; 1 denotes ethnic group “1” is significantly better at *p* ≤ 0.01; 2 denotes ethnic group “2” is significantly better at *p* ≤ 0.01.

**Table 4 ijerph-17-03391-t004:** Individual skills pre vs. post: mean (SD).

Ethnicity	Run			Jump			Stationary Dribble			Catch			Kick		
	Pre	Post	∆	Pre	Post	∆	Pre	Post	∆	Pre	Post	∆	Pre	Post	∆
White	10 (2) ^2^	10 (2)	0.49	6 (2)	8 (3)	1.71	2 (3) ^2^	3 (3)	1.01	8 (4)	10 (3)	1.656	8 (3)	9 (3)	1.27 *
South Asian	8 (4)	11 (1)	3.01 **	6 (3)	7 (3)	1.47	1 (2)	6 (3)	4.39 *	8 (2)	10 (2)	2.216	7 (3)	13 (2) ^2^	6.07 **
	Throw			Roll			Sprint (Secs)			Long-Jump (cm)			Medicine Ball Throw (cm)		
	Pre	Post	∆	Pre	Post	∆	Pre	Post	∆	Pre	Post	∆	Pre	Post	∆
White	6 (3) ^1^	6 (3)	−0.05	8 (2)	4 (2)	−3.74 **	3.63 (0.46)	3.29 (0.39)	–0.3	63 (17)	83 (18) ^1^	20.36 **	109 (23) ^1^	120 (33) ^1^	11 ^1^
South Asian	4 (2)	9 (1) ^2^	4.84 **	7 (2)	7 (4) ^2^	0.14	3.80 (0.84)	3.23 (0.59)	–0.5	68 (26)	65 (20)	–2.51	84 (46)	84 (24)	–0.17

* denotes significant within-subject changes at *p* ≤ 0.01; ** denotes significant within-subject changes at *p* < 0.0001; ^1^ denotes ethnic group “1” is significantly better at *p* ≤ 0.01; ^2^ denotes ethnic group “2” is significantly better at *p* ≤ 0.01; *p* ≤ 0.01 ∆ (delta) denotes mean difference.

**Table 5 ijerph-17-03391-t005:** Percentage of children at poor, near mastery, and mastery for each movement process characteristic of the skill at pre and post.

.	South Asian (*n* = 18)						White (*n* = 21)						Ethnic Comparison	
	PRE (%)			POST (%)			PRE (%)			POST (%)			PRE	POST
**Skill/movement process characteristic from CMSP**	**0**	**1**	**2**	**0**	**1**	**2**	**0**	**1**	**2**	**0**	**1**	**2**	**X^2^ (V)**	**X^2^ (V)**
Run C1—Arms move in opposition to legs, elbows bent	44	28	28	17	11	72	10	14	76	29	10	62	10.2 ** (0.50 ^##^)	0.78 (0.14)
Run C2—Brief period of suspension: both feet off the ground	11	6	83	0	6	94	0	0	100	0	0	100	4.9 (0.31)	1.60 (0.18)
Run C3—Narrow foot placement; lands on heel or toe; not flat footed	17	28	56	6	0	94	29	5	67	10	19	71	4.4 (0.33)	5.88 (0.33)
Run C4—Length of stride even; path of movement horizontal	11	6	83	0	0	100	5	0	95	0	0	100	2.2 (0.22)	NA
Run C5—Non-support leg flexed to approximately 90 degrees	22	11	67	0	11	89	24	14	62	10	19	71	0.1 (0.06)	2.5 (2.50)
Run C6—Eyes focused forward	11	6	83	0	0	100	0	5	95	0	0	100	2.5 (0.25)	NA
Jump C1—Preparatory: flexion of both knees; arms behind body	61	17	22	17	11	72	67	14	19	24	24	52	0.1 (0.06)	1.8 (0.21)
Jump C2—Arms extend forcefully; forward and upward to full extension above the head	61	11	28	39	11	50	29	43	29	14	29	57	6.2 * (0.39 #)	3.9 (0.31)
Jump C3—Take-off and landing on both feet simultaneously	17	22	61	11	0	89	0	10	90	10	19	71	6.8 * (0.38)	5.3 (0.31)
Jump C4—Take-off on both feet simultaneously; landing non-simultaneuous	56	22	22	100	0	0	95	5	0	76	19	5	10.6 ^++^ (0.48 ^##^)	6.8 * (0.36)
Jump C5—Arms move downward during landing	39	17	44	22	11	67	19	33	48	5	14	81	2.4 (0.25)	2.7 (0.266)
Jump C6—Balance maintained on landing	6	6	89	6	0	94	29	33	38	24	10	67	11.5 ^++^ (0.52 ^##^)	5.7 (0.35)
Stationary dribble C1—Arm action independent of trunk	83	6	11	6	0	94	48	5	48	48	14	38	6.6 * (0.40 #)	15.8 ^++^ (0.59 ^##^)
Stationary dribble C2—Ball contacted with one hand at about belt/waist height	72	11	17	50	6	44	67	14	19	24	29	48	0.2 (0.06)	5.1 (0.35)
Stationary dribble C3—Pushes ball with fingertips (does not slap at ball with flat hand)	94	6	0	6	0	94	86	0	14	62	19	19	5.3 (0.31)	26.2 ^++^ (0.76 ^##^)
Stationary dribble C4—Ball contacts surface in front of or to the outside of foot on preferred side	56	11	33	11	0	89	57	0	43	14	33	52	3.3 (0.26)	10.6 ^++^ (0.45 #)
Stationary dribble C5—Controls ball for four consecutive bounces; feet not moved to retrieve ball	100	0	0	94	6	0	86	0	14	90	5	5	3.93 * (0.27)	1.3 (0.15)
Catch C1—Preparatory: hands in front of body; elbows flexed	33	22	44	0	0	100	14	10	76	10	5	86	4.2 (0.33)	3.9 (0.27)
Catch C2—Arms extend toward ball as it moves closer	0	11	89	0	0	100	10	5	86	0	5	95	2.9 (0.24)	1.3 (0.15)
Catch C3—Ball caught cleanly with hands/fingers (2)	78	11	11	78	6	17	52	33	14	43	14	43	3.3 (0.28)	5.1 (0.35)
Catch C4—Ball trapped against body/chest (1)	22	33	44	28	6	67	33	33	33	57	10	33	0.74 (0.14)	4.4 (0.33)
Catch C5—Ball tracked consistently and close to point of contact	6	0	94	0	0	100	19	0	81	10	10	81	1.7 (0.20)	5.3 (0.31)
Catch C6—Does not turn head/close eyes as ball approaches	6	0	94	0	0	100	24	10	67	10	14	76	5.7 (0.35)	6.8+ (0.35)
Kick C1—Rapid and continuous approach to ball	67	17	17	17	0	83	38	24	38	14	14	71	3.4 (0.29)	3.9 (0.27)
Kick C2—Elongated stride or leap immediately prior to ball contact	33	33	33	0	0	100	76	5	19	71	10	19	8.9 * (0.46 #)	33.0 ^++^ (0.81 ^##^)
Kick C3—Non kicking foot placed even with or slightly in back of ball	28	17	56	0	0	100	0	10	90	5	10	86	9.7 ** (0.45 #)	3.9 (0.27)
Kick C4—Leg swing is full; full backswing and forward swing of leg	61	6	33	0	0	100	62	10	29	71	14	14	0.28 (0.08)	36.6 ^++^ (0.86 ^##^)
Kick C5—Backswing coordinated with forward action of non kicking leg	100	0	0	0	0	100	95	0	5	5	38	57	1.26 (0.15)	13.5 ^++^ (0.51 ^##^)
Kick C6—Ball contacted with instep of kicking foot (shoe-laces) or toe	0	6	94	0	0	100	0	0	100	5	19	76	1.6 (0.18)	6.8+ (0.36)
Kick C7—Kicks through ball; leg action does not stop at ball contact	17	6	78	0	6	94	10	10	81	0	14	86	0.6 (0.12)	0.8 (0.14)
Overarm Throw C1—Wind-up initiated by downward movement of hand/arm	22	11	67	0	0	100	0	10	90	10	14	76	6.9 * (0.37)	6.8+ (0.36)
Overarm Throw C2—Hip and shoulder rotated so that non throwing side faces target	83	6	11	0	0	100	62	10	29	62	14	24	2.3 (0.24)	29.8 ^++^ (0.77 ^##^)
Overarm Throw C3—Steps (weight transferred) onto foot opposite throwing arm	78	17	6	67	11	22	76	14	10	86	5	10	0.2 (0.10)	1.9 (0.23)
Overarm Throw C4—Differentiated trunk rotation (2)	100	0	0	94	0	6	80	10	10	90	0	10	5.3 (0.31)	0.22 (0.07)
Overarm Throw C5—Block trunk rotation (1)	56	6	39	6	0	94	52	14	33	38	10	52	0.9 (0.15)	10.0 ^++^(.47 ^##^)
Overarm Throw C6—Timing of release/flight of ball appropriate (late release = downward flight; early release = upward flight)	39	11	50	0	0	100	19	19	62	19	33	48	2.0 (0.23)	17.4 ^++^ (0.58 ^##^)
Roll C1—Ball arm/hand swings down/back of trunk; chest/head face forward	78	11	11	0	0	100	38	10	52	0	14	86	8.3 * (0.44 #)	1.1 (0.27)
Roll C2—Arm action in vertical plane	28	0	72	0	0	100	0	0	100	86	0	14	8.6 ** (0.41 #)	36.6 ^++^ (0.86 ^##^)
Roll C3—Foot opposite ball hand strides forward toward cones	89	11	0	94	0	6	90	5	5	81	14	5	1.8 (0.19)	3.9 (0.27)
Roll C4—Bends knees; lowers body	6	0	94	0	0	100	24	0	76	33	38	29	2.7 (0.25)	26.8 ^++^ (0.73 ^##^)
Roll C5—Ball held in fingertips	39	11	50	0	0	100	5	10	86	86	0	14	7.9 * (0.43 #)	36.6 ^++^ (0.86 ^##^)
Roll C6—Ball released close to floor; bounces less than 4 inches high	11	22	67	0	6	94	10	14	76	29	33	38	0.49 (0.11)	16.5 ^++^ (0.59 ^##^)

** South Asian sig < white *p* < 0.001, * South Asian sig < white *p* < 0.05, ^++^ South Asian sig > white *p* < 0.001, + South Asian sig > white *p* < 0.001, ^##^ sig effect *p* < 0.001, # sig effect *p* < 0.05.

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
