# Peer review of "The Effects of Combined Movement and Storytelling Intervention on Motor Skills in South Asian and White Children Aged 5–6 Years Living in the United Kingdom"

_ijerph, 2020, doi:10.3390/ijerph17103391_

Round 1

Reviewer 1 Report

Review of: The effects of combined movement and storytelling on motor skills in South Asian and white children aged 5-6 years living in the UK

General comments:

The purpose of the study was to examine the effects of an intervention involving motor skills and storytelling in young children in the UK, with a specific focus on the differential effects between South Asian and white children. While the topic is certainly of interest to motor development researchers, there are several major issues that need to be addressed.

Storytelling is sometimes spelled as one word and sometimes hyphenated. It doesn’t matter which option is used, but please be consistent throughout.

In the results, the two groups are referred to as “ethnic group 1” and “ethnic group 2.” This is not only confusing, but seemingly dismissive of each of the groups. For clarity purposes, please revise the results (and methods where appropriate) section to rename “ethnic group 1” and “ethnic group 2” with their respective names. This should be done in the tables as well.

The primary issue with this manuscript are the repetitive and pervasive errors in grammar, syntax, and punctuation. While I have made suggestions to correct many of these, the entire manuscript needs to be edited for consistency in grammar, syntax, and punctuation. These errors detract from the significance and overall findings of the study.

One pervasive issue with the research design is the repeated use of the word “deprived” when referring to South Asian children. It is suggested throughout the manuscript that the South Asian children are at a great socioeconomic advantage in comparison to their white counterparts, but no such measure of SES or “depravity” was ever taken. Either the language throughout with regard to the SES and depravity of the South Asian children needs to be removed, a measure of SES needs to be included, or this needs to be mentioned as a major limitation in the study.

Specific comments:

Abstract:

Line 17: “interventions” should be singular

Lines 19-20: Syntax is off with this sentence. Suggest putting the list of motor skills in parentheses.

Introduction:

Line 31: There is actually a more recent edition of the textbook that is referenced. Suggest referencing the 8th edition (lead author Goodway).

Line 32: The semicolon should appear after the [2] reference and the comma should appear after “as such” (…by the age of 6 years [2]; as such, these skills are a key focus…)

Line 32: “year’s” should be “years’”

Lines 37-38: This is a run-on sentence. Put the information after groups in parentheses with a comma after e.g.

Line 38: Should read “…despite the ethnically diverse and multicultural societies…”

Line 41: apostrophe not needed in Asians

Line 44: commas are not needed after “considering” or “et al’s”

Line 45: physical activity has been abbreviated up to this point. Continue abbreviating throughout.

55: “at risk” should be “at-risk”

Line 61: “skills” and “interventions” should both be singular and “has” should be “have”

Line 62: “development” should be “developmental”

Line 66: comma not needed after “includes”

Line 75: comma is needed after [11-12] and the word “and” is needed after “language, “

Line 77: Suggest starting new paragraph with the sentence “There is some”

Line 77-78: This is a run-on sentence. Put the “i.e., storytelling” in parentheses. See comment in general comments about the spelling of storytelling.

Line 81: Remove the word “from” (the instance between groups and experience)

Lines 86-87: This sentence is confusing. Suggest simplifying to read: The current study sought to be the first study to: 1) Compare…

Line 88: remove “for whom little data exists” and replace the period with a semicolon

Line 90: remove “for which no data exists”

Materials and methods:

Line 93: Previously, the phrase “motor skill performance” was used. Please be consistent throughout.

Line 106: Only a 44.8% retention rate? This should be mentioned in the limitations.

Line 123: This statement should be clarified. It’s fair to compare the CMSP to the TGMD-2 (but not the TGMD), however it’s not fair or correct to say that it provides a more sensitive assessment of motor competence. Rather, the CMSP is a motor skill assessment tool that is based off the TGMD-2, but is more suitable for pre-school aged children.

Line 130: “has” should be “have”

Line 138: If you choose to capitalize Object Control and Total Test, capitalize Locomotor also

Lines 141-143: This is a run-on sentence. It’s not needed to state what the individual skills are again.

Line 150: “component” should be “components”; there should be “which is” following ‘near mastery’ – same after ‘poor’

Line 155: the word “and” is needed before medicine ball – and why is medicine ball capitalized?

Line 165: teaching does not need to be capitalized

Line 168: the phrase “motor development intervention” is redundant here. Just say movement and storytelling intervention.

Lines 173-177: This sentence is entirely unorganized and one big run-on sentence. Please rewrite entirely without run-ons, and use correct punctuation.

Line 179: End the sentence with academic groups. Start a new sentence with “When children”

Line 184: the word “and” is needed before “key words”

Line 186: the word “and” is needed before “this was”

Line 188: reword to “whereby this design previously resulted in increases in motor skill proficiency…”

Line 190: the word “and” is needed before “discuss key instruction”

Line 192: Run-on sentence. Put the information after e.g., in parentheses

Table 1: “Closing” should be capitalized in the title row; the bullet points in the fifth column are off

Line 221: the word “and” is needed before “where”

Results:

In the first paragraph under results, the skills are listed in italics, which makes it clearer to differentiate between the statistical information and the skill itself. Please follow suit in paragraph two under results.

Lines 248-249: Report which ethnic group rather than saying group 1 and group 2.

Table 2: Ethnicity should be indicated by name and not number. Why weren’t t-tests reported on differences in age, height, sitting height, weight, BMI, and wc between ethnicities? It is stated in the abstract that there were group differences at baseline. T-tests should be run and reported on this information.

Table 3: Same comment as above with ethnicity 1 and 2. Use the names.

Section 3.2: Individual motor skills. This information is all quite redundant. There must be a way to condense this information so that so much similar information per skill is not reported.

Table 4: Why is the entire table in italics?

Lines 317-338: Again, it seems that this information could be better summarized in a table of sorts. It’s very redundant and difficult to follow.

Line 323: South Asian needs to be capitalized

Line 337: South needs to be capitalized

Table 5: While this table provides good information, if there is a way to incorporate the actual statistics that were run (and discussed in lines 317-388), that should be done. Without doing that, I’m not sure this table is necessary. Also South Asian need to be capitalized in the top row and the table doesn’t need to be entirely in italics.

Discussion:

Line 343: South needs to be capitalized

Line 345: Comma needed after differences

Line 354: exists should be exist

Line 357: South needs to be capitalized

Lines 356-359: This is a run-on sentence.

Lines 360-365: It really isn’t necessary to explain each individual component here as that was already explained in the results and in a table. If this information is going to stay, it at least needs to be punctuated throughout.

Line 368: remove motor fitness factors from the example; as well needs to be two words

Line 370: South needs to be capitalized

Lines 367-371: This discussion is pretty weak. If you’re going to go into such an in depth discussion previously in the paragraph about the individual component differences, these sentences need to be expanded upon. What, specifically, do physical and mechanical factors of the specific skill components you mentioned have to do with the results of the South Asian children?

Line 375-378: This sentence is confusing and difficult to follow. Revise.

Lines 378-382: This is a run-on sentence.

Lines 402-404: This is a run-on sentence.

Lines 404-406: I have no idea what this sentence is saying. Revise.

Lines 406-423: All of this information really belongs in the introduction as it isn’t discussed in reference to the current findings. It provides background information and reasoning for the storytelling approach, which is highly appropriate for the introduction, but not the discussion section.

References:

#3: There is something missing in this reference.

Reviewer 2 Report

This paper addresses and important topic and is well conducted, the literature review is appropriate. I have only minor comments, and one major concern.

Beginning with the latter, I am a little worried that your findings can be explained by a regression to the mean effect. Please address this point.

Second, I would like to see some corrections for alpha, given that you have many individual comparisons the danger of Type I is high.

Third, the results can be difficult to follow with so many item-level comparisons, poor labelling in tables (e.g., ethinic group 1, proficiency level 0, why not label these for what they are - "South Asian").

I would appreciate a smaller paragraph on p.2. Also language is mentioned a fair bit, but then not analysed.

Finally, I am not entirely convinced as to why these two samples should be studied. In terms of fine motor skills, the opposite has been predicted see:

Luo, Z., Jose, P. E., Huntsinger, C. S., & Pigott, T. D. (2007). Fine motor skills and mathematics achievement in East Asian American and European American kindergartners and first graders. British Journal of Developmental Psychology, 25(4), 595–614. https://doi.org/10.1348/026151007X185329

Reviewer 3 Report

This paper addresses a new area in the development of FMS by combining a motor skills intervention for children in the early years stage of education with storytelling. This would be attractive to educators, as no additional time for physical activity needs to be made available in the curriculum. This study involved participants from deprived backgrounds, and compared the baseline level and rate of improvement in FMS in those from the South Asian ethnic minority group to white children. It was established that FMS performance at baseline was poorer among children from South Asian background. After the intervention, this group outperformed those from a white background. The combination of motor skills and storytelling in this intervention was effective at improving FMS in 5-6-year-old children. A time-by-ethnicity effect was established, with greater gains in FMS for those from South Asian background than those from white background.

·       Major attention needs to be paid to the second half of the introduction. The penultimate paragraph contains information that is not clearly linked, and several sentences do not flow – partly due to incorrect punctuation. New concepts and theoretical models underpinning this intervention are introduced in the discussion, and would be better placed in the introduction.

·       The results section would make more logical sense if results at baseline (difference by ethnicity) were presented first, followed by results at post-test/intervention results.

·       The discussion doesn’t include the strengths and limitations of this study, and does not make clear suggestions for future research based on the findings/limitations of the current study.

·       The discussion does not provide possible explanations of why particular skills or skill components were more or less sensitive to improvement (or specifically in a certain ethnic group) as a result of this intervention.

·       Grammar and punctuation require attention throughout. Sections/sentences requiring attention have been highlighted yellow. Particular attention needs to be given to the use of linking words (thus, hence, …), commas, apostrophes, and consistency of capitalisation of south Asian (or South Asian if you prefer) and white (or White).

·       Several sentences would benefit from being split into two or more concise and clear sentences.

·       Specific points not related to grammar, punctuation, or sentence structure are addressed below.

Introduction

p.2 line 46 and 54. What is meant by ‘these groups at higher risk’?

Is it those with low FMS?

Is it those who are south Asian?

Is it those with low SES?

p.2 line 46 “Preliminary findings suggest…” Preliminary findings of what? What is the context? What are these studies you refer to? Just having the reference is not enough to let me follow the argument.

p.2 line 62. “evidencing that…” disadvantaged pre-schoolers’ results cannot be generalised to those with developmental delays. Did those in the study described have developmental delays?

p.2 line 73. This sentence needs to be more specific. Are you talking about all children? Those with poor motor skill also doing poorly in language skills? Those with low SES? Those form minority backgrounds?

p.2 line 89. “and their associated behavioural components” No description is given of this concept, nor is any mention made of behavioural components in the method or results, but this is mentioned again in the discussion.

Materials and methods

p.3-4 line 131-132 and line 141-146. Please re-write sentences/sections to make sense.

p.4 line 149-151. What constitutes ‘correct performance’?

p.4 line 160-162. It would be helpful if you could state a sentence or two on how the medicine ball throw was conducted: how many trials were completed, furthest counts/average score, etc.

p.4 line 167. If you have groups of 25-30, and 87 participants, that makes 3 groups. Yet, in line 168, you write ‘both groups’.

p.5 line 222. Please include the values for weak, moderate, and strong association for Cramer’s V.

Results

p.5 line 230. Which skill process does ‘bounce’ correspond with? This term has not been used in any previous description.

p.6 table 2. A comparison (t-test) of difference between the ethnic groups on each variable would be helpful to determine if the background characteristics were similar.

SES or a marker of deprivation should also be included in these characteristics.

Please include a note of which ethnicity corresponds to which number group.

p.6 line 245. “time and ethnicity” Should this be ‘time-by-ethnicity interaction’?

If so, it would be helpful to have the main (within person) effect of time, the main (between person) effect of ethnicity, and the interaction effect reported, rather than just the interaction effect.

This is repeated in all individual skill results reported.

BF interpretation does not need repeating in each skill report.

p.6 Table 3. Given the pre- and post-data for ‘locomotor’ and ‘object’, I am surprised to see how the ‘7-skill’ improvement from pre to post for ethnicity 2 can be so large. From the methods section it seems that the 7-skill is the combination of the locomotor and object?

The values in table 3 do not seem to correspond with values in table 4.

p.7 line 282-285. What is the difference in the two between-ethnicity results reported? This post-hoc test does not add value to the result of the ANOVA as it is a main effect, so no interaction needs disentangling. Unless the post-hoc test reveals which of the two groups scored higher, this does not need reporting.

p.7-8 line 308-313. See above regarding post-hoc test.

Which ethnic group is group 3? This is not mentioned anywhere else in the text.

p.10 line 323 & 324. “higher levels” – a larger proportion?

Discussion

p.11 line 344-345. “this study is one of only a few” One of only a few in what?

p.11 line 347-349. This is an unnecessary repetition of a statement in lines 216-220 in the methods section.

p.11-12 line 353-359 and 361-365. Please re-write to make sense.

p.12 line 360. “behaviour component level” How does this differ from the skill process and skill product measures you have been talking about? Why is this ‘behaviour level’?

Section 4.1 introduces a large number of new concepts not previously discussed in this paper (introduction or related to methods). These require greater explanation/definition/conceptualisation in the context.

p.12 line 370-371. The conclusion presented here is not warranted by your data, or can be exclusively drawn from the information presented in the previous sentences.

p.12 line 389 “narrowing the ethnic gap” You narrowed the motor skills gap.

p.12 line 399-400. “Furthermore, …to motor development” This model has not been previously introduced and it is unclear what you refer to here. What constraints? How were they structured?

p.12 line 402-406. Re-write to make sense.

p.12-13 line 406-416. This is the foundation on which you’ve built your intervention and needs to be mentioned in the introduction. You can then relate to it again later in the discussion. It also needs a clarification how this embodied approach benefits the motor development in specific. It currently explains how the language skill development would benefit from the addition of movement, rather than the other way around. Or, if you aim to demonstrate that adding the storytelling doesn’t detract from the motor development, but can be more easily accommodated in the curriculum due to the language skill benefits, this needs to be made explicit.

p.13 line 416 onwards. Make it clear that you are now discussing strengths, limitations, and options for future research.

p.13 line 431. “In line with this…” The jump results for White children are not in line with the throw action description in the previous sentence.

p.13 line 435-439. Re-write to make sense.

Conclusion

Consider re-phrasing the conclusion into several shorter sentences, so that your key points come through more clearly.

p.13 line 447-448. This is not the future research suggestion you mention in earlier sections of the discussion, and seems entirely unrelated to the rest of the article.

Round 2

Reviewer 1 Report

I commend the authors for such a thorough and precise revision of the paper. With the exception of a few minor grammatical/syntax issues in the newly added information, I believe the manuscript is now suitable for publication. I presume the editorial office will handle the proofreading. 

Nice work!